# Fast Monte Carlo Rendering via Multi-Resolution Sampling

Qiqi Hou*          Zhan Li†          Carl S Marshall‡          Selvakumar Panneer§          Feng Liu¶

Portland State University          Intel          Intel          Portland State University

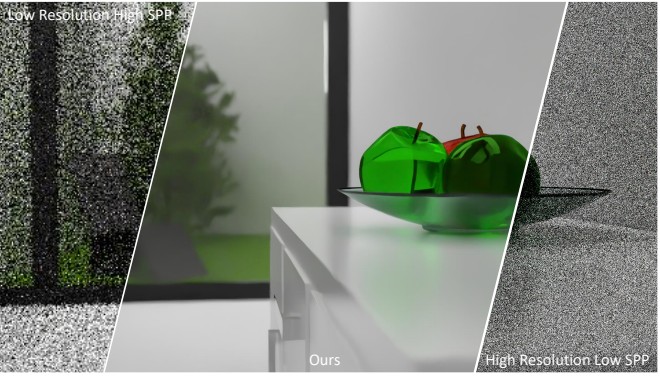 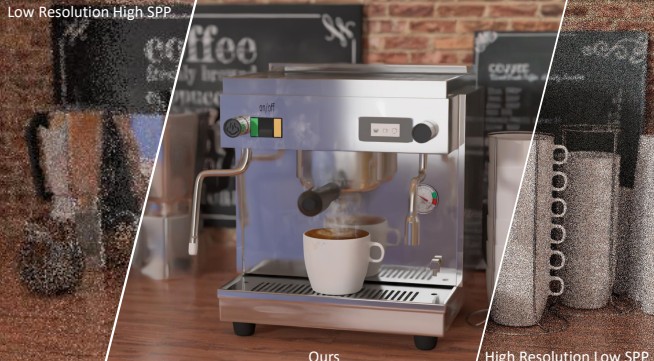

Figure 1: Our fast hybrid rendering method takes a low resolution with a high sample rate rendering (LRHS) and a high resolution with a low sample rate rendering (HRLS) as inputs, and produces the high resolution high quality result.

## ABSTRACT

Monte Carlo rendering algorithms are widely used to produce photo-realistic computer graphics images. However, these algorithms need to sample a substantial amount of rays per pixel to enable proper global illumination and thus require an immense amount of computation. In this paper, we present a hybrid rendering method to speed up Monte Carlo rendering algorithms. Our method first generates two versions of a rendering: one at a low resolution with a high sample rate (LRHS) and the other at a high resolution with a low sample rate (HRLS). We then develop a deep convolutional neural network to fuse these two renderings into a high-quality image as if it were rendered at a high resolution with a high sample rate. Specifically, we formulate this fusion task as a super resolution problem that generates a high resolution rendering from a low resolution input (LRHS), assisted with the HRLS rendering. The HRLS rendering provides critical high frequency details which are difficult to recover from the LRHS for any super resolution methods. Our experiments show that our hybrid rendering algorithm is significantly faster than the state-of-the-art Monte Carlo denoising methods while rendering high-quality images when tested on both our own BCR dataset and the Gharbi dataset [14][1].

**Index Terms:** Computing methodologies—Computer graphics—Ray tracing

## 1 INTRODUCTION

Physically-based image synthesis has attracted considerable attention due to its wide applications in visual effects, video games, design visualization, and simulation [25]. Among them, ray tracing methods have achieved remarkable success as the most practical realistic image synthesis algorithms. For each pixel, they cast numerous rays that are bounced back from the environment to collect

---

*e-mail: qiqi2@pdx.edu
†e-mail: lizhan@pdx.edu, equal contribution
‡e-mail: carl.s.marshall@intel.com
§e-mail: selvakumar.panneer@intel.com
¶e-mail: fliu@pdx.edu

[1]https://github.com/hqqxyy/msspl

photons from light sources and integrate them to compute the color of that pixel. In this way, ray tracing methods are able to generate images with a very high degree of visual realism. However, obtaining visually satisfactory renderings with ray tracing algorithms often requires casting a large number of rays and thus takes a vast amount of computations. The extensive computational and memory requirements of ray tracing methods pose a challenge, especially when running these rendering algorithms on resource-constrained platforms and impede their applications that require high resolutions and refresh rates.

To speed up ray tracing, Monte Carlo rendering algorithms are used to reduce ray *samples per pixel* (spp) that a ray tracing method needs to cast [10]. For instance, adaptive reconstruction methods control sampling densities according to the reconstruction error estimation from existing ray samples [57]. However, when the ray sample rate is not sufficiently high, the rendering results from a Monte Carlo algorithm are often noisy. Therefore, the ray tracing results are usually post-processed to reduce the noise using algorithms like bilateral filtering and guided image filtering [27,37,42,44,48,51,56]. Recently, deep learning-based denoising approaches are developed to reduce the noise from Monte Carlo rendering algorithms [2,9,23,28]. These methods achieved high-quality results with impressive time reduction, and some of them are incorporated into commercial tools, such as VRay Renderer, Corona Renderer, and RenderMan, and open source renderers like Blender. However, real-time ray tracing is still a challenging problem, especially on devices with limited computing resources.

Our idea to speed up ray tracing is to reduce the number of pixels that we need to estimate color values. For instance, upsampling by $2 \times 2$ can reduce 75% of pixels that need ray tracing to estimate color. There are two main challenges in super-resolving a Monte Carlo rendering. First, it is still a fundamentally ill-posed problem to recover the high-frequency visual details that are missing from the low-resolution input. Second, a Monte Carlo rendering is subject to sampling noise, especially when it is produced at a low spp rate. Upsampling a noisy image will often amplify the noise level as well. To address these challenges, we propose to generate two versions of rendering: a low-resolution rendering but at a reasonable high spp rate (LRHS) and a high-resolution rendering but at a lower spp rate (HRLS). LRHS is less noisy while the more noisy HRLS can potentially provide high-frequency visual details that are inherently difficult to recover from the low resolution image.

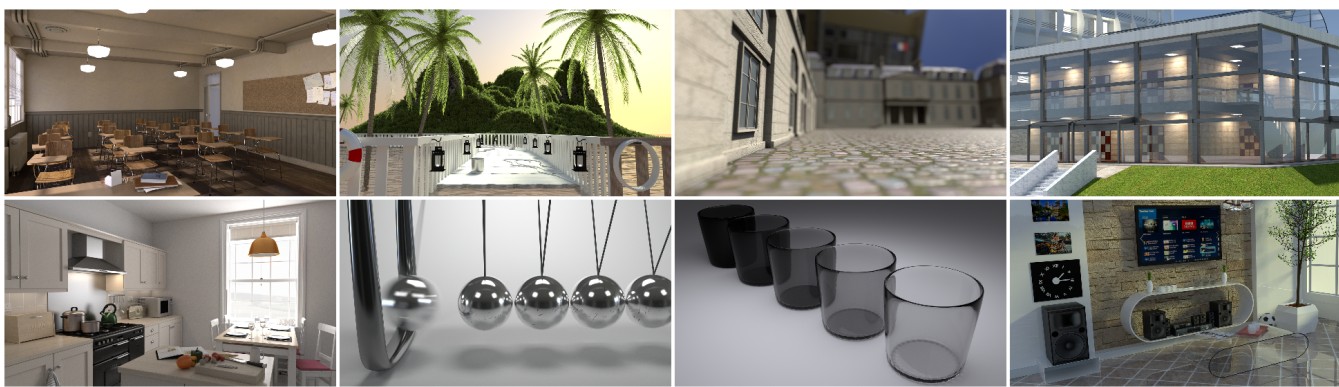

Figure 2: Examples from our BCR dataset.

We accordingly develop a hybrid rendering method dedicated for images rendered by a Monte Carlo rendering algorithm. Our neural network takes both LRHS and HRLS renderings as input. We use a de-shuffle layer to downsample the HRLS rendering to make it the same size as LRHS and to reduce the computational cost. Then we concatenate the features from both LRHS and HRLS and feed them to the rest of the network to generate the high-quality high resolution rendering. Our experiments show that given the hybrid input, our method outperforms the state-of-the-art Monte-Carlo rendering algorithms significantly.

To train our network, we collected a large Blender Cycles Ray-tracing dataset, which contains 2449 high-quality images rendered from 1463 models. The dataset consists of various factors that affect the Monte Carlo noise distribution, such as depth of field, motion blur, and reflections. We render the images at a range of spp rates, including 1-8, 12, 16, 32, 64, 250, 1000, and 4000 spp. All the images are rendered at the resolution of 1080p. Each image contains not only the final rendered result but also the intermediate render layers, including albedo, normal, diffuse, glossy, and so on.

This paper contributes to the research on photo-realistic image synthesis by integrating Monte Carlo rendering and image super resolution for efficient high-quality image rendering. First, we explore super resolution to reduce the number of pixels that need ray tracing. Second, we use multi-resolution sampling to both reduce noises and create visual details. Third, we develop a large ray-tracing image dataset, which will be made publicly available.

## 2 RELATED WORK

Monte Carlo rendering is an important technology for photo-realistic rendering. It aims to reduce the number of rays that a ray tracing algorithm needs to cast and integrate while synthesizing a high quality image [10, 22]. Conventional Monte Carlo rendering algorithms investigate various ways to adaptively distribute ray samples [8, 13, 20, 32, 38–41, 46, 47]. When only a small number of rays are casted, the rendered images are often noisy. They are typically filtered using various algorithms [11, 21, 29, 30, 36, 42–44, 49]. Due to the space limit, we refer readers to a recent survey on Monte Carlo rendering [57].

Our research is more related to the recent deep learning approaches to Monte Carlo rendering denoising. Kalantari *et al.* trained a multilayer perceptron neural network to learn the parameters of filters before applying these filters to the noisy images [23]. Bako *et al.* extended this method by employing filters with spatially adaptive kernels to denoise Monte Carlo renderings [2]. They developed a convolutional neural network method to estimate spatially adaptive filter kernels. Chaitanya *et al.* developed an encoder-decoder network with recurrent connections to denoise a Monte Carlo image sequence [9]. Recently, Kuznetsov *et al.* [28] developed a deep convolutional neural network approach that combines adaptive sampling and image denoising to optimize the rendering

performance. Different from the above methods, Gharbi *et al.* argued that splatting samples to relevant pixels is more effective than gathering relevant samples for each pixel for denoising. Accordingly, they developed a novel kernel-splatting architecture that estimates the splatting kernel for each sample, which was shown particularly effective when only a small number of samples were used [14]. Compared to these methods, our method improves the speed of Monte Carlo rendering by reducing the number of pixels that we need to cast rays for.

Our work also builds upon the success of deep image super resolution methods [1, 12, 15, 19, 26, 33–35, 50, 52, 55]. Dong *et al.* developed the first deep learning approach to image super resolution [12]. They designed a three-layer fully convolutional neural network and showed that a neural network could be trained end to end for super resolution. Since that, a variety of neural network architectures, such as residual network [16], densely connected network [18], and squeeze-and-excitation network [17], are introduced to the task of image super resolution. For instance, Kim *et al.* developed a deep neural network that employs residual architectures and obtained promising results [26]. Lim *et al.* further improved super resolution results by removing batch norm layers and increasing the depth of networks [34]. Zhang *et al.* developed a residual densely connected network that is able to explore intermediate features via local dense connections for better image super resolution [54]. Zhang *et al.* recently reported that a channel-wise attention network which is able to learn attention as guidance to model channel-wise features could more effectively super resolve a low resolution image [53]. While these image super resolution methods achieved promising results, recovering visual details that do not exist in the input image is necessarily an ill-posed problem. Our method addresses this fundamentally challenging problem by leveraging a high-resolution image but rendered at a low ray sample rate. Such an auxiliary rendering can be quickly rendered and yet provide visual details that do not exist in the low resolution input rendered at a high sample rate.

## 3 THE BLENDER CYCLES RAY-TRACING DATASET

We develop a Blender Cycles Ray-tracing dataset (BCR) that consists of a large number of high quality scenes together with the ray-tracing images and the intermediate rendering layers. We will share BCR with our community.

### 3.1 Source Scenes

Blender's Cycles is a popular ray tracing engine that is capable of high-quality production rendering. It has an open and active community where thousands of artists share their work. Using the Blender community assets, we collected over 8000 scenes under Creative Commons Licenses, which allow us to share our dataset with the research community. We rendered these scenes at 4000 spp and manually checked the rendered images and all the rendering

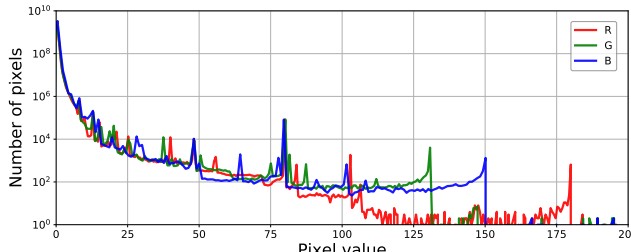

Figure 3: Pixel value distribution of our BCR dataset. The rendered images use the *scene linear color space* and the pixel value is represented in Float32. We use the logarithmic scale for the *y* axis. While 99.98% pixels are in the range of [0, 10], the distribution has a long tail. For a better visualization, we only show the pixel value in the range of [0, 200].

layers. We eliminated scenes with missing materials, lack of high frequency information, or with noticeable rendering noises even rendered at 4000 spp. This culling process reduced the total number of source scenes to 1465. These remaining scenes produced 2449 images by rendering from 1 to 10 viewpoints per scene. We split the dataset into 3 subsets: 2126 images from 1283 scenes as the training set, 193 images from 76 scenes as the validation set, and 130 images from 104 scenes as the test set. There is no overlap scene among them. As shown in Figure 2, our dataset covers various optical phenomena, such as motion blur, depth of field, and complex light transport effects. It covers a variety of scene contents, including indoor scenes, buildings, landscapes, fruits, plants, vehicles, animals, glass, and so on.

## 3.2 Rendering Settings

To generate the high-quality "ground-truth" renderings, we rendered each scene at 4000 spp. As described previously, we noticed that the rendered images for some scenes still contain noticeable noises even when rendered at 4000 spp and we removed them through manual inspection. On average, it took around 20 minutes to render an image on an Nvidia Titan X Pascal GPU. We set the rendering resolution to $1920 \times 1080$ or $1080 \times 1080$ to cover the most content of scenes. For each image, we provide both the final rendered image and the render layers, which are essential for Monte Carlo rendering [2, 3, 9, 23, 24, 28, 31, 32]. In total, each image has 33 rendering layers, including albedo, normals, depth, diffuse color, diffuse direct, diffuse indirect, glossy color and so on. Please refer to our project website for more details. All images in the BCR dataset can be produced using the render layers as follows [6]:

$$I_{HR} = I_{Diff} + I_{Gloss} + I_{Sub} + I_{Trans} + I_{Env} + I_{Emit}, \quad (1)$$

where the diffuse, gloss, subsurface, trans layers can be generated with their color, direct light and indirect light layers

$$
\begin{aligned}
I_{Diff} &= I_{DiffCol} * (I_{DiffDir} + I_{DiffInd}), \\
I_{Gloss} &= I_{GlossCol} * (I_{GlossDir} + I_{GlossInd}), \\
I_{Sub} &= I_{SubCol} * (I_{SubDir} + I_{SubInd}), \\
I_{Trans} &= I_{TransCol} * (I_{TransDir} + I_{TransInd}).
\end{aligned}
\quad (2)
$$

Besides rendering 4000spp images as ground truth, we rendered each scene at 1-8, 12, 16, 32, 64, 128, 250, and 1000 spp as input for Monte Carlo rendering enhancement algorithms, including ours. The rendered images and the auxiliary results in the scene were saved in the *scene linear color space*, which closely corresponds to natural colors [5]. These images were rendered with a high dynamic range. The pixel values were represented in Float32. As shown in Figure 3, 99.98% pixel values were in the range of [0, 10]. However, the pixel value distribution had a long tail. We also noticed that many of the very large values come from the firefly rendering artifacts. Therefore we removed these outliers by clipping at value 100. An

| Dataset | Images | Scenes | SPP | Layers |
|---|---|---|---|---|
| Kalantari [23] | 500 | 20 | 4, 8, 16, 32, 64, 32000 | 5 |
| KPCN [2] | 600 | - | 32, 128, 1024 | 6 |
| Chaitanya [9] | - | 3 | 1, 4, 8, 16, 32, 256, 2000 | 3 |
| Kuznetsov [28] | 700 | 50 | 1, 2, 4, 8, 16, 1024 | 4 |
| **BCR dataset** | **2449** | **1463** | 1-8, 12, 16, 32, 64, 128, 250, 1000, 4000 | **33** |

Table 1: Monte Carlo rendering dataset comparison.

image in the scene linear space can be converted to sRGB space for visualization in this paper as follows.

$$
s =
\begin{cases}
0 & \text{if } l \leq 0, \\
12.92 \times l & \text{if } 0 < l \leq 0.0031308, \\
1.055 \times l^{\frac{1}{2.4}} - 0.055 & \text{if } 0.0031308 < l < 1, \\
1 & \text{if } l \geq 1,
\end{cases}
\quad (3)
$$

where $l$ and $s$ indicate the pixel value in scene linear color space and sRGB respectively [4].

## 3.3 Low Resolution Image Generation

A straightforward way to generate low resolution images is to change the output resolution in Cycles. However, directly rendering a low resolution image does not always work [7]. For example, some scenes are modelled using a subdivision technology and changing the rendering resolution will disrupt the inherent relationship among the material and geometry settings in the scene files and thus cause mismatch between images rendered at different resolutions. Therefore, we generate low resolution images by downsampling the corresponding high resolution rendered images via the nearest neighbour degradation. We did not use bilinear or bicubic sampling as the nearest neighbor degradation more accurately simulates a real-world rendering engine. That is, rays for low resolution renderings are sampled at a sparse grid compared with high resolution ones.

## 3.4 Monte Carlo Rendering Dataset Comparison

We compare our dataset with those used in recent deep learning-based Monte Carlo rendering denoising algorithms, including [2, 9, 23, 28]. As reported in Table 1, our dataset has over $3\times$ the amount of images and over $25\times$ the number of scenes than the other datasets. Moreover, most these existing datasets are private and we will make our dataset public.

## 4 METHOD

Our method takes a low-resolution-high-spp image $I_{LRHS}$ and its corresponding high-resolution-low-spp image $I_{HRLS}$ as input and aims to estimate a corresponding HR image $I_{SR}$. $I_{LRHS}$ contains the RGB channel, while $I_{HRLS}$ is composed of RGB channel and extra layers, including Albedo, Normal, Diffuse, Specular, Variance layer as these extra layers can provide high-frequency visual details.

As shown in Figure 4, we design a two-encoder-one-decoder network to estimate the HR image. Given $I_{LRHS}$ and $I_{HRLS}$, our network firstly extracts the features $F_{LRHS}$ and $F_{HRLS}$, respectively. We leverage a downscale module with deshuffle layers [45] instead of pooling layers to downscale the feature maps as deshuffle layers can keep the high-frequency information. Compared with upscaling LRHS features, downsampling HRLS features to the same size of $F_{LRHS}$ can reduce the computational complexity of the network significantly. It also enables the features to fuse in the earlier layer of the network. We obtain the fused feature $F_0$ by combining $F_{HRLS}$ with $F_{LRHS}$ through a fusion module and feed it to a sequence of residual dense groups (RDG) [53, 54]. With the feature $F_G$ from RDGs, we combine it with $F_{LRHS}$ by element-wise adding. Finally, we upscale the resulting dense feature $F_{DF}$ and predict the final HR image $I_{SR}$ through a convolutional layer. Below we describe the network in detail.

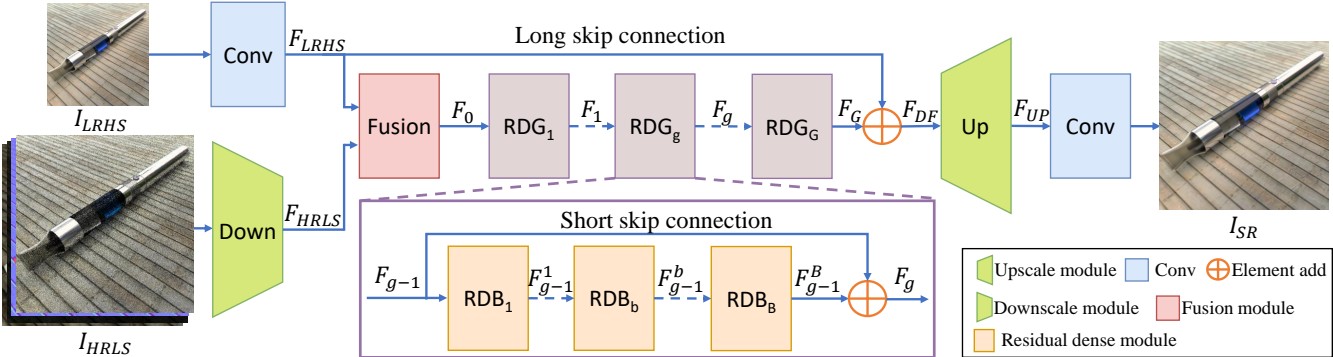

Figure 4: The architecture of our network. Our network takes a low-resolution-high- spp rendering (LRHS) and its corresponding high-resolution-low-spp rendering (HRLS) as input and predicts the final high-resolution-high-quality image.

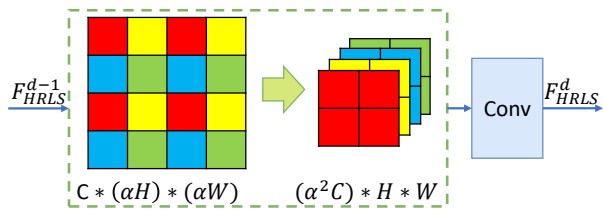

Figure 5: Deshuffle layer for downscaling feature maps.

**LRHS shallow feature** $F_{LRHS}$. Following [34, 53, 54], we adopt a convolutional layer to get the shallow feature $F_{LRHS}$

$$F_{LRHS} = H_{lrhs}(I_{LRHS}), \quad (4)$$

where $H(\cdot)$ indicates the convolution operation.

**HRLS shallow feature** $F_{HRLS}$. We first extract the shallow feature from $I_{HRLS}$ with a convolutional layer,

$$F_{HRLS}^0 = H_{hrls}(I_{HRLS}). \quad (5)$$

Inspired by ESPCN [45], we design a deshuffle layer to downscale the features. As shown in Figure 5, we downscale the feature map with a stride of $\alpha$. In our network, we set $\alpha = 2$. To downscale the feature map, we stack deshuffle layers together. Supposing our network has $D$ deshuffle layers, we can get the output $F_{HRLS}$

$$F_{HRLS} = DSF^D(DSF^{D-1}(\cdots DSF^1(F_{HRLS}^0)\cdots)), \quad (6)$$

where $DSF(\cdot)$ indicates the operation of the deshuffle layer. By downscaling auxiliary features, our network can work in the size of the LRHS image, which can significantly reduce the computational complexity of the overall network.

We concatenate $F_{LRHS}$ from LRHS image and $F_{HRLS}$ from HRLS into a combined feature map $F_0$.

**Residual densely connected block.** We employ the densely connected network and residual groups to build the backbone of our neural network as they are shown effective for image super resolution [53, 54]. In our network, we use 4 convolutional layers in each residual densely connected block (RDB). By stacking $B = 5$ RDBs, we build a residual densely connected group (RDG) as follows,

$$F_g = RDB_B(RDB_{B-1}(\cdots RDB_1(F_{g-1})\cdots)) + F_{g-1} \quad (7)$$

We predict the dense feature $F_{DF}$ with $G = 3$ RDGs as follows,

$$F_{DF} = RDG_G(RDG_{G-1}(\cdots RDB_1(F_0)\cdots)) + F_0 \quad (8)$$

**Upscale.** In our network, we adopt the shuffle layer from ES-PCN [45] to upscale the features and estimate the high resolution prediction $I_{SR}$,

$$I_{SR} = H_{Rec}(UP(F_{DF})), \quad (9)$$

where $UP(\cdot)$ indicates the operation of upscale [45].

**Loss function.** The BCR dataset is in the scene linear color space. As shown in Figure 3, the pixel value distribution of this BCR dataset has a long tail. $\ell_1$ loss cannot handle it well because it might be biased to the extremely large pixel values. To handle this problem, we adopt the following robust loss

$$\ell_r = \frac{1}{N} \sum_{p \in I_{HR}} \frac{|I_{HR}^p - I_{SR}^p|}{\beta + |I_{HR}^p - I_{SR}^p|}, \quad (10)$$

where $\beta$ indicates the robust factor. For the small difference, $\ell_r$ works quite similarly to $\ell_1$. For the extremely large difference, $\ell_r$ will be close to but always below 1. This will prevent our network from the bias towards rare but extremely large pixel values. We set $\beta = 0.1$ in our experiments.

**Implement details.** We set the kernel size of all convolutional layers to $3 \times 3$, except for the fuse convolutional layer, whose kernel size is $1 \times 1$. Every convolutional layer is followed by a RELU layer, except for the last convolutional layer. The shallow features, fusion features, and dense features have 64 channels. During each iteration of the training, we randomly select the spp of $I_{HRLS}$ from the set of [1-8, 12, 16, 32] and the spp of $I_{LRHS}$ from the set of [2-8, 12, 16, 32, 64, 128, 250, 1000, 4000] while making sure that the spp of $I_{HRLS}$ is smaller than that of $I_{LRHS}$.

We use PyTorch to implement our network. We use a mini-batch size of 16 and train the network for 500 epochs. It takes about one week on one Nvidia Titan Xp for training. We use the SGD optimizer with the learning rate of $10^{-4}$. We also perform data augmentation on-the-fly by randomly cropping patches. In order to save data loading time, we pre-crop training HR images into $300 \times 300$ large patches. During training, we further crop smaller patches on those large patches. The final patch size of HR is set to 96 for $\times 2$, 192 for $\times 4$, and 256 for $\times 8$. We select the model that works best on the validation set.

## 5 EXPERIMENTS

We evaluate our method by comparing it with representative state-of-the-art denoising methods for Monte Carlo rendering and image super resolution algorithms. We also conduct ablation studies to further examine our method. We use two metrics to evaluate our results. First, we adopt RelMSE (Relative Mean Square Error) to report the results in the *scene linear color space*, which is defined as

$$RelMSE = \lambda_1 * \frac{(I_{SR} - I_{HR})^2}{I_{HR}^2 + \lambda_2}, \quad (11)$$

where $\lambda_1 = 0.5$ and $\lambda_2 = 0.01$ when experimenting on our BCR dataset following KPCN [2]. For the Gharbi dataset, we use the evaluation code from its authors [14] where $\lambda_1 = 1$ and $\lambda_2 = 10^{-4}$.

We also use PSNR to evaluate the results in the *sRGB* space. For our BCR dataset, we convert images to sRGB to calculate PSNR use Equation 3. For the Gharbi dataset, we convert images to the sRGB space using codes provided by its authors as follows [14],

$$s = min(1, max(0, l)), \quad (12)$$

| Method | 2spp | | 4spp | | 8spp | |
|---|---|---|---|---|---|---|
| | PSNR | RelMSE | PSNR | RelMSE | PSNR | RelMSE |
| Input | 18.12 | 0.2953 | 21.51 | 0.1400 | 24.75 | 0.0646 |
| KPCN [2] | 25.87 | 0.0390 | 27.31 | 0.0299 | 28.11 | 0.0276 |
| KPCN-ft [2] | 31.03 | 0.0078 | 33.69 | 0.0043 | 35.83 | 0.0026 |
| Bitterli [3] | 26.67 | 0.0293 | 27.22 | 0.0252 | 27.45 | 0.0226 |
| Gharbi [14] | 30.73 | 0.0068 | 31.61 | 0.0057 | 32.29 | 0.0050 |
| Ours×2 | (4 - 1) | | ( 8 - 2) | | (16 - 4) | |
| | 33.27 | 0.0044 | 35.15 | **0.0027** | **36.74** | **0.0019** |
| Ours×4 | (16 - 1) | | (32 - 2) | | (64 - 4) | |
| | **33.94** | **0.0039** | 35.21 | 0.0028 | 36.31 | 0.0022 |
| Ours×8 | (64 - 1) | | (128 - 2) | | (250 - 4) | |
| | 31.37 | 0.0075 | 32.35 | 0.0057 | 33.14 | 0.0049 |

Table 2: Comparison on our BCR dataset. Ours ×2 indicates that our method performs x2 super resolution and (4 - 1) indicates that our method takes 4 spp LRHS and 1 spp HRLS as input, which is effectively 2 spp on average for all the pixels.

| Method | 4 spp | | 8 spp | | 16 spp | |
|---|---|---|---|---|---|---|
| | PSNR | RelMSE | PSNR | RelMSE | PSNR | RelMSE |
| Input | 19.58 | 17.5358 | 21.91 | 7.5682 | 24.17 | 11.2189 |
| Sen [44] | 28.23 | 1.0484 | 28.00 | 0.5744 | 27.64 | 0.3396 |
| Rousselle [41] | 30.01 | 1.9407 | 32.32 | 1.9660 | 34.36 | 1.9446 |
| Kalantari [23] | 31.33 | 1.5573 | 33.00 | 1.6635 | 34.43 | 1.8021 |
| Bitterli [3] | 28.98 | 1.1024 | 30.92 | 0.9297 | 32.40 | 0.9640 |
| KPCN [2] | 29.75 | 1.0616 | 30.56 | 7.0774 | 31.00 | 20.2309 |
| KPCN-ft [2] | 29.86 | 0.5004 | 31.66 | 0.8616 | 33.39 | 0.2981 |
| Gharbi [14] | 33.11 | **0.0486** | 34.45 | **0.0385** | 35.36 | **0.0318** |
| Ours×2 | (8 - 2) | | (16 - 4) | | (32 - 8) | |
| | **34.02** | 1.5025 | **35.30** | 1.4902 | **36.43** | 1.4748 |
| Ours×4 | (32 - 2) | | (64 - 4) | | (128 - 8) | |
| | 33.94 | 5.5586 | 35.22 | 5.6781 | 35.97 | 5.7436 |
| Ours×8 | (128 - 2) | | (16 - 8) | | (32 - 16) | |
| | 31.56 | 3.7228 | 32.60 | 4.2300 | 33.22 | 4.5045 |

Table 3: Comparison on the Gharbi dataset [14].

where $l$ indicates images in the scene linear space, $s$ indicates images in the sRGB space.

## 5.1 Comparison with Denoising Methods

We compare our method to both state-of-the-art traditional denoising methods, including Sen *et al.* [44], Rousselle *et al.* [41], Kalantari *et al.* [23], Bitterli *et al.* [3], and recent representative deep learning based methods, including KPCN [2] and Gharbi *et al.* [14]. Unlike other methods, our method takes both a LRHS rendering and a HRLS rendering as input. Therefore, we compute the average spp for our input as $spp_{avg} = spp_{LRHS}/s^2 + spp_{HRLS}$, where $s$ indicates the super resolution scale. For instance, in Table 2, "Ours ×2" indicates that our method performs ×2 super resolution and (4 - 1) indicates that our method takes 4 spp LRHS and 1 spp HRLS as input, which is effectively 2 spp on average. We conducted on the comparisons on both the Gharbi dataset and our BCR dataset.

Table 2 compares our method to Bitterli *et al.* [3], KPCN [2], and Gharbi *et al.* [14]. We used the code / model shared by their authors in this experiment. For KPCN [2], we provide another version of the results produced by their neural network but fine-tuned on our BCR dataset. This experiment shows that our method, especially ours ×2 and ×4, outperform the state-of-the-art methods by a large margin. Specifically, our ×4 method wins 2.91dB on PSNR and 0.0039 on RelMSE when the spp is 2. When spp is relatively high, our ×2 method wins 0.91dB on PSNR and 0.0007 on RelMSE. Figure 7 shows several visual examples on the BCR dataset. Our results contain fewer artifacts than the other methods.

We were not able to compare to additional methods on our BCR

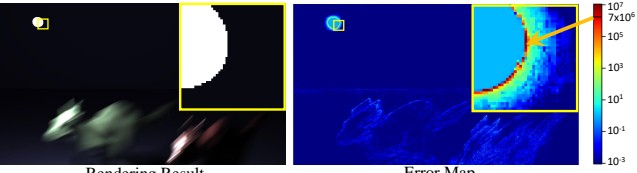

Figure 6: Error map visualization.

| spp | 4 | 8 | 16 | 32 | 64 | 128 |
|---|---|---|---|---|---|---|
| Rousselle [41] | | | | | | 13.3 |
| Kalantari [23] | | | | | | 10.4 |
| Bitterli [3] | | | | | | 21.9 |
| KPCN [2] | | | | | | 14.6 |
| Sen [44] | 281.2 | 638.1 | 1603.1 | 4847.8 | - | - |
| Gharbi [14] | 6.0 | 10.1 | 18.9 | 35.9 | 67.0 | 156.5 |
| Ours×2 | | | | | | 0.362 |
| Ours×4 | | | | | | 0.118 |
| Ours×8 | | | | | | 0.052 |

Table 4: Comparison of runtime cost (second) to denoise a 1024 × 1024 image. The data is from Gharbi [14]. If the runtime is constant, we report it in the last column. Our ×2, ×4 and ×8 method are at least 17×, 51× and 115× faster than the start-of-the-art methods, respectively.

dataset as these methods work with other rendering engines or use very different input formats. We compare with these methods on the Gharbi dataset, as reported in Table 3. We obtained the results for the comparing methods from Gharbi *et al.* [14]. For our results, we directly used our neural network trained on our BCR dataset without fine-tuning it on the Gharbi training dataset.[2] As shown in Table 3, our method outperforms all the other methods in terms of PSNR.

However, the RelMSE of our results is higher than some of the existing methods, such as KPCN [2] and Gharbi [14]. We looked into the discrepancy between the results measured using PSNR and RelMSE. We found that the RelMSE metric is heavily affected by a small number of pixels with abnormally large errors in our results. Figure 6 shows the RelMSE error map of one of our results with a much larger error than Gharbi. Figure 6 shows an example of our result where the errors concentrate in the region around the bright light, with 16 pixels having errors larger than $10^6$, which contribute to most of the error of the whole image. After excluding these 16 pixels, while our error is still larger than Gharbi, the difference is much smaller. We would like to point out that our method was trained on our BCR dataset only and was not fine-tuned on the Gharbi dataset as its training set is not available. Moreover, BCR and the Gharbi dataset were rendered using different engines and thus contained different intermediate layers. To test on the Gharbi examples, we had to set the variance layer to a constant value, which compromises our results.

Figure 8 shows visual comparisons between our method and several existing methods. Although our RelMSE is higher than Gharbi [14], our results look more plausible, which is consistent with our higher PSNR values measured in the sRGB space. In the first example, the seat in our result contains much fewer artifacts. In the second example, the highlight in our result is more accurate than others. It shows that our model and BCR dataset have a great generalization capability.

**Speed and memory.** We report the speeds of the above methods in Table 4. We use the same setting as Gharbi [14] and obtain the timing data for the comparing methods from them as well. For our method, We report the aggregated spp for our method by combining samples used to render both HRLS and LRHS. Since all the methods use the

---

[2]We removed one image from the Gharbi testing set as its source model is also included in our BCR training set.

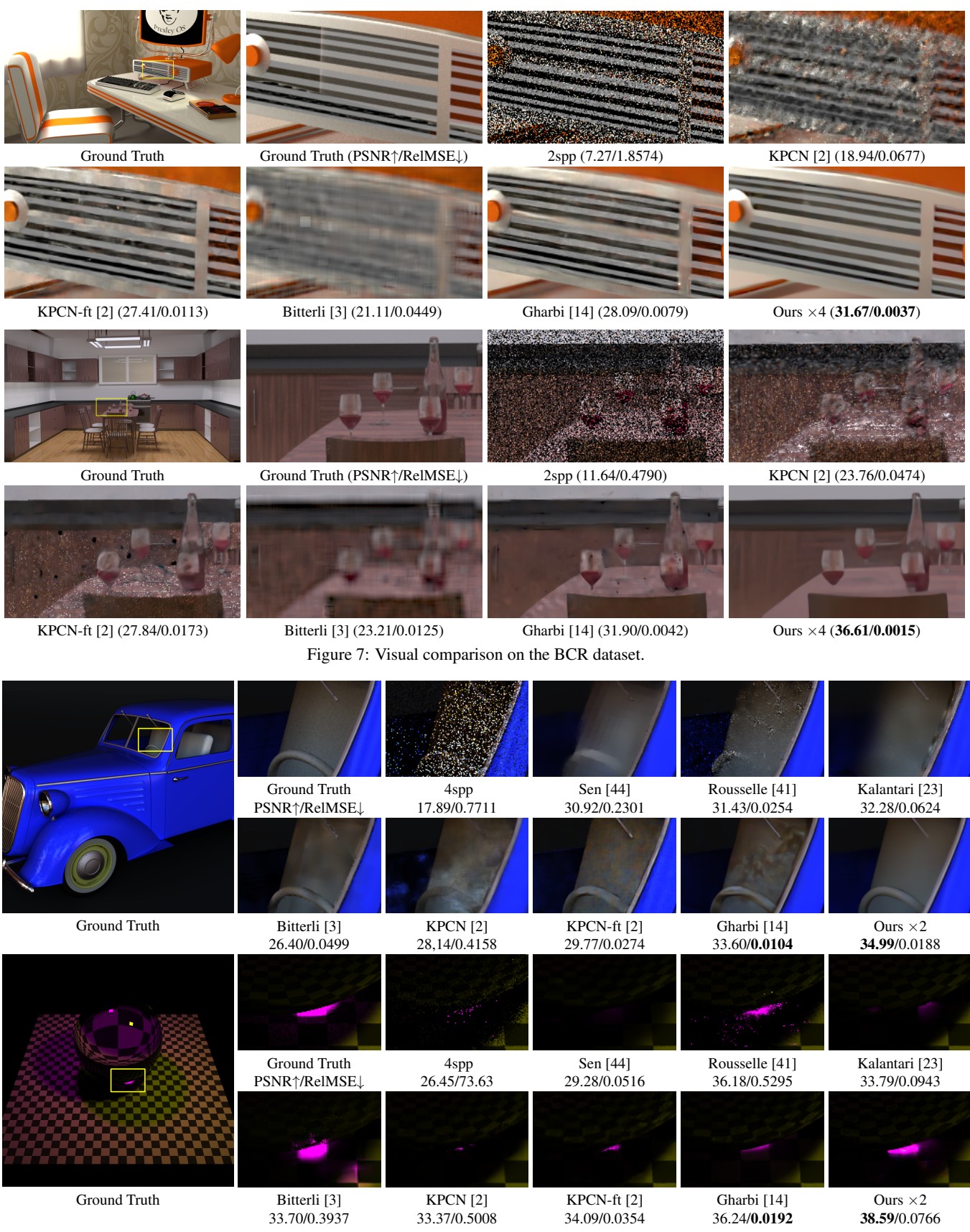

Figure 7: Visual comparison on the BCR dataset.

Figure 8: Visual comparison on the Gharbi dataset [14].

| Methods | ×2 | | ×4 | | ×8 | |
|---|---|---|---|---|---|---|
| | PSNR | RelMSE | PSNR | RelMSE | PSNR | RelMSE |
| Bicubic | 30.57 | 0.0141 | 25.39 | 0.0858 | 22.36 | 0.2473 |
| EDSR [34] | 32.01 | 0.0079 | 30.70 | 0.0119 | 27.97 | 0.0241 |
| RCAN [53] | 32.03 | 0.0084 | 30.73 | 0.0117 | 27.92 | 0.0253 |
| Ours | **38.40** | **0.0015** | **34.27** | **0.0039** | **31.08** | **0.0079** |

Table 5: Comparison with super resolution methods on the BCR dataset.

| HRLS / LRHS | 1spp | | 2spp | | 4spp | |
|---|---|---|---|---|---|---|
| | PSNR | RelMSE | PSNR | RelMSE | PSNR | RelMSE |
| 2spp | 32.14 | 0.0056 | - | - | - | - |
| 4spp | 32.94 | 0.0048 | 33.76 | 0.0038 | - | - |
| 8spp | 33.52 | 0.0042 | 34.41 | 0.0033 | 35.20 | 0.0027 |
| 16spp | 33.94 | 0.0039 | 34.88 | 0.0030 | 35.71 | 0.0025 |
| 32spp | 34.22 | 0.0037 | 35.21 | 0.0028 | 36.06 | 0.0023 |
| 64spp | 34.42 | 0.0035 | 35.44 | 0.0027 | 36.31 | 0.0022 |
| 128spp | 34.56 | 0.0035 | 35.60 | 0.0026 | 36.49 | 0.0021 |

Table 6: The effect of spp values on the final rendering results.

same spp, we only include the time needed for denoising. We report the time of processing one $1024 \times 1024$ image on one Nvidia Xp GPU. We can find that our ×2, ×4 and ×8 methods are at least 17×, 51× and 115× faster than the state-of-the-art method Gharbi [14]. In addition, our ×2, ×4 and ×8 network models require peak GPU memories of 1134 MB, 749 MB, and 737 MB process a $1024 \times 1024$ image respectively.

## 5.2 Comparisons with Super Resolution Methods

We also compare our method with several baseline methods that use super resolution to upsample the low-resolution-high-spp renderings to the target size. In this experiment, we used the trained models shared by the authors of these super resolution methods [34, 53] and fine-tuned them on our BCR dataset. As reported in Table 5, our method generates significantly better results than these super resolution methods. While this comparison is unfair to these baseline methods, it indeed shows the benefits of taking an extra high-resolution-low-spp rendering as input. As shown in Figure 11, our results contain more fine details that are missing from the super resolution results.

## 5.3 Ablation study

We now examine several key components of our method.
**Input layers of $I_{LRHS}$ and $I_{HRLS}$.** We examine how the rendering layers affect the final results. In this experiment, we use 1 spp for $I_{HRLS}$ and 4000 spp for $I_{LRHS}$. The upsampling scale is set to ×4. Our neural network contains two input branches, one for $I_{LRHS}$ and the other for $I_{HRLS}$. In this experiment, we fix the input layer of one branch to RGB while changing the input layers of the other. For the model with "None", we remove this branch. As shown in Figure 9, compared with no inputs, $I_{HRLS}$ can greatly improve the results. Among various input layers, RGB improves the results by a large margin. The result can be further improved if the $I_{HRLS}$ takes all rendering layers. We believe that these improvements come from the high frequency information in the $I_{HRLS}$. For the $I_{LRHS}$, while all the input layers still help, the RGB result alone can achieve the best result. We conjecture that since $I_{LRHS}$ is rendered with a high spp, its RGB layer is already of very high quality, and the other intermediate layers do not further contribute. On the other hand, the intermediate layers for $I_{HRLS}$ provide useful information for denoising, which is consistent with the findings of the previous denoising methods [2, 14].
**Robust loss $\ell_r$.** We examine the effect of the parameter $\beta$ in our robust loss. We also compare it to the standard $\ell_1$ loss. In this experiment, we use 4000 spp for $I_{LRHS}$ and 1 spp for $I_{HRLS}$. The

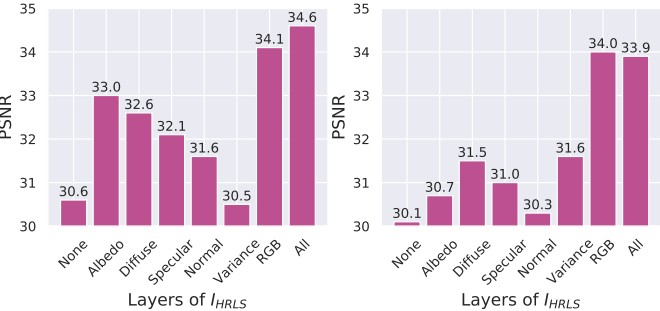

Figure 9: The effect of input rendering layers.

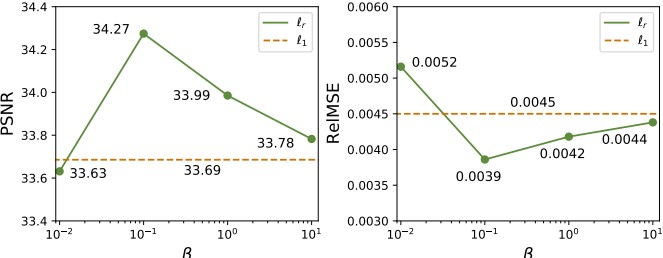

Figure 10: Comparison between $\ell_r$ and $\ell_1$.

upsampling scale is set to ×4. The input channels of $I_{LRHS}$ and $I_{HRLS}$ are set to RGB. Figure 10 shows that the robust loss $\ell_r$ with $\beta = 0.1$ outperforms $\ell_1$ by a large margin as it can avoid the bias towards a very small number of pixels with extremely large pixel values. We also find that using a too large or too small $\beta$ will be harmful to the results. This is because a very large $\beta$ value reduces the robust loss to the $\ell_1$ loss while a very small beta value makes the loss always close to 1 without regard to the error between the output and the ground truth.
**SPP of $I_{LRHS}$ and $I_{HRLS}$.** We examine how our method works with different spp values used to render $I_{LRHS}$ and $I_{HRLS}$. In the experiment, we set the upsampling scale to ×4. Table 6 shows rendering at high spp values consistently leads to better final results.

## 6 CONCLUSION

This paper presented a hybrid rendering method to speed up Monte Carlo rendering algorithms. We designed a two-encoder-one-decoder network for this task. Our network takes a low resolution image with a high spp and a high resolution image with a low spp as inputs and estimates the high resolution high quality images. We built a large-scale ray-tracing dataset Blender Cycles Ray-tracing dataset. Our experiments showed that our method is able to generate high quality high resolution images quickly. Our experiments also showed that HRLS and the robust loss are helpful to generate high quality results.

### ACKNOWLEDGMENTS

The source models in Figure 1, 2, 4, 7, and 11 are used under a Creative Commons License from kujaba, darkst0ne, cczero, LukeLiptak, Christophe Seux, nickbrunner, samytichadou, MarcoD, Jay-Artist, jgilhutton, Oldfrizt, Ndakasha, GyngaNynja, racingfoli, Kless and PepSu. The source models in Figure 6 and 8 are from Gharbi [14]. This project is supported by a gift from Intel.

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
