# OpenReview forum: "Fast Monte Carlo Rendering via Multi-Resolution Sampling"
_graphicsinterface.org/Graphics_Interface/2021/Conference/Second_Cycle — GI 2021_

### Official Review · Reviewer_CAoJ · 2021-05-03
**This paper describes a speedup for MC rendering by generating a low resolution, high sample per pixel image, and a high resolution, low sample image, and ultimately fusing them using a deep convolutional neural network.**

**Rating:** 8
**Confidence:** 4

**Review:**

The paper is interesting and clearly demonstrates the methods and results. It explains sufficient details of the different parts of the methods, as well as how the training was performed. The reproducibility appear to be quite good.

This paper builds upon two main ideas:
1) additional channels for more information
2) RDGs for efficient super-resolution.

It compounds them with their original idea of using two different resolutions at different sample counts.

The resulting method seem to produce excellent results, consistently improving error metrics when they make sense.
The images show very clean and smooth gradients where other techniques usually produce artifacts. Edges of objects are quite sharp, and the shape is preserved, even though the technique seem to wash out some high-frequency textures. It also has trouble with details too small for the LRHS but too hard to sample for the HRLS. This is noticeable on the Figure 11, the reflection does not follow the curve. Even though, the output is very clean and does not represent a hard failure case. It still visibly outperforms current methods.

Another major contribution is the creation and public release of a large dataset of reference images with layer information. This will hopefully allow future research to use a common ground to compare new methods, and not only for machine learning.

The Ablation study is interesting. I would like to dig deeper into it. RGB is by far the most important channel. Starting with only RGB, how many channels should we add to obtain, say, 99% of the quality, i.e., how much of these layers can we shave off? How much of an improvement can we expect? Also there is no list of all layers, likely for space reasons. I do not think it is really needed, but I hardly imagine that every layer brings relevant information. I see that the HRLS image is down sampled to match the one of the LRHS, but maybe a dimensionality reduction technique could further reduce it by combining layers in a more information dense way?

The paper starts by raising concerns about memory requirements on resource-constrained platforms, but never actually states how the presented method fares in that regard. The input is at least 230MB with all the layers, which is far from free. A few words on maximum memory consumption would be appreciated.

---

### Official Review · Reviewer_PyoN · 2021-05-03
**This paper combines images of high and low resolution sampled respectively with low and high spp. using a convolutional neural network. It generates high quality noise free images from noisy Monte-Carlo images and compares favorably to previous work.**

**Rating:** 8
**Confidence:** 3

**Review:**

This paper proposes a denoising algorithm for Monte-Carlo rendering that combines
a high resolution image with low sampling rate with a low resolution image with high
sampling rate, using a neural network. The network combines downsampling of the high resoluton image,
a set of convolutional layers, and a up-sampling step to produce the result. The papers shows
impressive results and systematic (and favorable) comparisons to existing work.

The idea to combine high and low resolution images is interesting but immediately raises the question: why
is that any better than sampling the same total amount of rays on the high resolution image? I understand that
low image frequencies are probably better reconstructed with the low resolution image. An attempt to explain
this (in a discussion?) would help clearing out any doubts.

Path tracing values can easily vary in a range from 1 to 10^5 (in particular when outdoor lighting is involved).
Figure 3 shows a much more limited range of pixel values, which is rather surprising. I also do not
understand the sudden significant drops for the red/green/blue channels at values 130,150 and 180. Is this
graph built on images that already suffered a tone-mapping step?

It is not very clear in the paper whether we're talking about general path tracing or direct illumination only.
This should be cleared up right away in the introduction.

As described by Figure 4, the algorithm down-samples the high resolution low ssp. image right away. In this case
it is hard to figure out what the high resolution image is really useful to in the whole process. I suppose that
most high frequency features do not come from this image but from the associated buffers. As such, if high resolution
features only come from light (e.g. in a caustic), the proposed algorithm would not be any better than traditional
denoising.

Also in Figure 4, the to input images have rather different colors. Is that a tone-mapping problem?

The release of the path-traced scene database itself is a major asset in this paper, that will certainly be used a lot by future work.

Comparisons in Section 5.1 and 5.2 are generally well done and convinced me that the paper adds on to
previous work and is therefore worth publishing.

---

### Official Review · Reviewer_ijA3 · 2021-05-04
**Best-in-class rendering denoising and super-resolution for low sample settings using a combination of a high-resolution low samples and low-resolution high samples as input into a two-encoder-one-decoder neural network. This comes alongside a novel, to be publicly released, Blender Cycles rendering dataset at different sample settings. The quality of the manuscript and contributions are sufficiently notable and articulate enough for publication.**

**Rating:** 8
**Confidence:** 4

**Review:**

The work makes use a neural network (two-encoder-one-decoder) approach to combine together low-resolution high quality setting renderings with high-resolution low quality renderings with the goal of outputting the upscaled super-resolution rendering results at the improved final image quality. While the combination of a low-resolution rendering prior to guide the settings for the final full-resolution image rendering is already pretty common with production renderers across the industry, the additional twist of using a neural network to achieve a similar results here, brings a certain level of originality.  There are sufficient details regarding the network architecture, data processing and training to deem the work reproducible. The approach compares favorably to prior work, in recent years, both performance-wise and image quality-wise as demonstrated through the inference speed, PSNR and relMSE metric comparisons and through some visual image comparisons. The authors will also publicly share a sizeable curated novel dataset of Blender renderings (with additional image features layers) at different target quality settings for the benefit of the community. The stated contributions are appreciable, both for the rendering denoising and the super-resolution domains, and the manuscript provides sufficient information and clarity to conceivably merit publication.

Comments:
* It is not immediately clear what the full list of “33 image layers” are in the rendered data – it might help to elaborate that in the text or in an Appendix.
* The choice of $\beta = 0.1$ for the robust loss function could use additional justification to support the particular choice.
* There are several sizeable rendering datasets available in the wild. Some of them are in fact used later in this submission for tabulated comparisons. The repeated mentions that such datasets do not exist might thus be somewhat misleading and should be elaborated upon (One such example is the Disney dataset which is available alongside the KPCNN paper: https://studios.disneyresearch.com/2017/07/20/kernel-predicting-convolutional-networks-for-denoising-monte-carlo-renderings/).
* Denoise/super-resolution networks tend to be somewhat tuned for a particular renderer and might not generalize and thus not perform as well when used with output renderings from different renderers. The danger here is that by primarily using Blender Cycles data and its output feature channels, we are in effect training for performing this task with Cycles as the image generator. Analyzing how much this is the case and perhaps fine-tuning with rendering results from more diverse sources might be worthwhile pursuing (e.g. through the addition of Mitsuba or other commercial engine renderings).
* Generating the low-resolution rendering indirectly through processing of high resolution images instead of true native resolution renderings can have other complications depending on how the actual rendering engine, Cycles in this case, has been configured. It is not immediately obvious whether additional precautions have been taken while keeping this in mind.
* Is this approach (a) temporally stable and (b) does it produce deterministic filtered image results? Given that the neural network has not been trained on scene motion image sequences, it is rather unlikely that is the case. Admittedly, this might not be in scope for the particular work but should be considered in the future.
* It is not immediately obvious how well this pipeline handles transparencies, DoF, motion blur and reflections. While some limited image results are shared with the manuscript, it can be rather valuable if a larger set of image comparisons are made available, either through supplementary material or by making the codebase public alongside the dataset, so the community can perform a more detailed assessment of the work.
* Quality metrics can frequently be somewhat misleading since neither PSNR nor relMSE sufficiently capture the human perceptual quality of the output. The small set of included image results do show favorable comparisons of this work when compared with other prior work while at the same time, unsurprisingly, we still see various artifacts manifesting, such as loss of local detail, blurring and incorrectly extrapolated shading.
* Curating the dataset by eliminating renderings from BCR that did not converge even after 4000 spp can be biasing against challenging scenes which stand to benefit more from improved denoising.
* Both discarding apparent fireflies as well as clamping results introduces some degree of additional bias. The original pixel values can be many orders of magnitude higher than any threshold limit as seen in the histogram plot. Moreover, clipping values to 100 in the linear space, in a truly HDR scene e.g. where sun and sky and other sources are visible alongside shadows/dark regions in the rendering, significantly restricts the dynamic range. Perhaps this is one of the reasons the particular pipeline is struggling with input similar to what is seen in Figure 6.

Some other minor corrections/improvements for the manuscript:
* Introduction paragraph 2: typo "open source renders like Blender" -> "renderers"
* Experiments section (first sentence): “… comparing it representative state-of-the art denoising”- > "comparing it with representative"
* Section 5.1 Comparison with Denoising Methods: Second sentence typo correct  "ssp" -> "spp"
.  Second paragraph typo “This experiment showes that" -> "shows that"
* Loss function subsection: $l_1$ loss “can not handle”  -> “cannot handle”

---

### Meta-Review · Area_Chair_fr31 · 2021-05-06

**Recommendation:** Accept
**Confidence:** 5

**Metareview:**

Reviewers consistently acknowledged the novelty and well soundness of the proposed technique, as well as the quality of the results and the extent of the evaluation/comparison with respect to previously existing techniques. Furthermore, the release of the rendering data set is in itself a very valuable contribution to the field. There is no doubt that the paper should be accepted for publication.

---

### Decision · Program_Chairs · 2021-05-08

Accept